# Nomophobia and Self-Esteem: A Cross Sectional Study in Greek University Students

**DOI:** 10.3390/ijerph20042929

**Published:** 2023-02-08

**Authors:** Elissavet Vagka, Charalambos Gnardellis, Areti Lagiou, Venetia Notara

**Affiliations:** 1Laboratory of Hygiene and Epidemiology, Department of Public and Community Health, School of Public Health, University of West Attica, 11521 Athens, Greece; 2Department of Fisheries and Aquaculture, University of Patras, 30200 Messolonghi, Greece

**Keywords:** nomophobia, self-esteem, nomophobia questionnaire, Rosenberg’s self-esteem scale, university students, smartphone

## Abstract

Nomophobia is a relatively new term describing someone’s fear, discomfort, or anxiety when his/her smartphone is not available. It is reported that low self-esteem may contribute to an individual’s tendency for nomophobia. The aim of this particular study was to investigate the association between nomophobia and self-esteem among Greek university students. The study sample consisted of 1060 male and female university students aged 18 to 25 years, participating on a voluntary basis with an online anonymous questionnaire. Data were collected through “Nomophobia Questionnaire (NMP-Q)” and “Rosenberg’s self-esteem scale (RSES)”. All participants exhibited some level of nomophobia, with the moderate level prevailing (59.6%). Regarding self-esteem categories, 18.7% of the participants showed low self-esteem, while the rest showed normal/high levels. Students with low self-esteem were twice as likely to exhibit a higher level of nomophobia compared to those with normal/high (adj Cum OR = 1.99, *p* value < 0.001). Additionally, women and students having fathers without a university education had a higher risk of exhibiting a greater level of nomophobia (adj Cum OR = 1.56 and 1.44, respectively, *p* values ≤ 0.008). It was observed that low self-esteem and nomophobia are closely connected. Further investigation into this particular issue is needed to explore potential causality between them.

## 1. Introduction

The extended use of new technologies and virtual communications (i.e., personal computers, tablets, and mobile phones), for personal/private or vocational purposes, has resulted in changes in peoples’ everyday plans and behavior [1]. Individuals’ lives have been positively affected by the introduction of smartphones and new technologies due to advanced applications and opportunities such as telecommunication, socialization, online gaming, and digital learning. However, excessive use has been documented to have adverse effects [2]. 

One of these effects is called nomophobia. Nomophobia is a relatively new term describing someone’s fear, discomfort, or anxiety when a smartphone is not available. Additionally, it is characterized as a pathological distress when disconnected from technology [3]. It is the fear of being unable to make or receive phone calls, send or receive text messages, lose internet connections and access to social networking sites, or be unable to access online information [4]. This fear has been prevailing as a result of the numerous applications and communication capabilities computers and smartphones provide. It is stated that personality issues may interfere with this relationship. For instance, individuals who are self-sufficient and mindful are less prone to nomophobia, while those who prefer sacrificing and surrendering in relationships are more susceptible [5]. It is also observed that nomophobia has an impact on interpersonal relationships and interactions, causing a sense of separation and isolation from the real world [6]. Several studies examined nomophobia in relation to the development of mental and personality disorders [7], loneliness, happiness, and self-esteem [8], mainly in the younger population [9,10]. 

Self-esteem is described as a subjective assessment of an individual’s worth as a person [11]. For instance, self-esteem does not always reflect an individual’s skills and abilities. Additionally, while self-esteem has been defined as a sense of self-acceptance and self-respect, a strong sense of self-esteem does not always indicate that an individual thinks of himself or herself to be superior to others [12]. 

Self-esteem is frequently examined in relation to problematic smartphone use. Several psychological factors, including low self-esteem, loneliness, depression, and an extroverted personality may contribute to an individual’s tendency for nomophobia, but they may also be some of the consequences of it [13]. 

It is reported that low self-esteem is associated with dysfunctional smartphone use [14,15]. Smartphone overuse is influenced by the mimicry of others, social anxiety, and low self-esteem [16]. The desire for social approval, as reflected by the amount of time spent composing and reading messages, has also been linked to low self-esteem [17]. Individuals with low self-esteem stated that they use their smartphone excessively via indirect ways such as phone calls, texting, and e-mail communication [14,18], while others with high self-esteem prefer face to face communication [15]. Due to the fact that young people with low self-esteem feel more secure behind a screen, they are more often engaged in virtual reality through electronic devices [14]. According to the findings of a recent study conducted among 228 undergraduate university students in India, nomophobia was found to be significant negatively associated with self-esteem [19].

Despite the fact that the prevalence of nomophobia in university students has recently gained a lot of attention, there are relatively few studies exploring the relationship between nomophobia and self-esteem. In Greece, there is no scientific evidence on nomophobia among university students, neither on the risk factors nor the characteristics associated with it.

Therefore, the aim of this particular study was to investigate the association between nomophobia and self-esteem among Greek university students.

## 2. Materials and Methods

### 2.1. Subjects

The specific cross-sectional study included 1060 university students, 308 (29.1%) males and 752 (70.9%) females, more than half were aged between 18 to 20 years, and 25.9% were working. Parents’ educational background was subdivided into two groups (i.e., university graduated and others).

### 2.2. Study Procedure

The sample was recruited from six faculties of the University of West Attica located in Athens, the capital city of Greece. The questionnaire was anonymous and due to COVID restrictions it was distributed online during the lecture time. The study researcher provided all required information and was available online via the Micro soft Teams platform; data were obtained electronically.

### 2.3. Data Collection Tools

The data collection tool consisted of three parts including the (a) socio-demographic characteristics and smartphone use (i.e., age, gender, parents’ educational background, hours, calls, messages, and e-mails/day), (b) nomophobia questionnaire, and (c) the Rosenberg’s self-esteem scale.

#### 2.3.1. Nomophobia Questionnaire (NMP-Q)

The Nomophobia Questionnaire (NMP-Q) is a 20-item Likert scale with a 7-point range from 1 to 7, with 1 indicating ‘strongly disagree’ and 7 indicating ‘strongly agree’. The nomophobia total score is between 20 and 140 (20 absence, 21–59 mild, 60–99 moderate, and 100–140 severe). Additionally, NMP-Q consists of four dimensions: (a) Not being able to communicate, (b) Losing connectedness, (c) Not being able to access information, and (d) Giving up convenience [4]. The NMP-Q was adapted and validated in the Greek language [20]. Exploratory and confirmatory factor analysis revealed a four-factor structure (subscales) in agreement with the original one [4]. The total scale presented a high internal consistency, similar to the original NMP-Q (Cronbach alpha values are 0.945 for both questionnaires). Moreover, the Cronbach alpha values for each factor were for (a) 0.936, (b) 0.895, (c) 0.867, and (d) 0.854, close to those of the original NMP-Q, which were 0.939, 0.827, 0.819, and 0.874, respectively. 

#### 2.3.2. Rosenberg Self Esteem Scale (RSES)

The RSES is a ten-item scale that assesses global self-esteem by measuring both positive and negative thoughts about oneself. It is widely regarded as a reliable and accurate quantitative self-worth evaluation instrument. The Likert scale is scored on a range from 0 to 3, with 0 representing strongly disagree and 3 representing strongly agree; items 3, 5, 8, 9, and 10 are rated in reverse. An additive score of the Likert values less than 15 indicates low self-esteem, a score of 15–25 indicates normal, and a score of more than 25 indicates high. Initially, the test was intended to assess high school students’ self-esteem. Since its development, the scale was used in different age groups, including adults [12]. The Cronbach’s alpha of the study RSES was 0.81, suggesting a high internal consistency of the scale, similar to that found by Galanou et al. (2014) [21].

### 2.4. Data Analysis

Data analysis was undertaken by simple univariate techniques or by modeling the data through ordinal logistic regression. Categorical variables were presented as absolute and relative (%) frequencies. The associations between nomophobia categories and sociodemographic characteristics of participants were evaluated through *χ*^2^ for linearity. Continuous variables were given by their mean and median values, while comparisons between them, due to the skewed distributions, were evaluated through the Kruskal–Wallis test.

An ordinal logistic regression model was developed, having as a dependent variable the total nomophobia score classified into three escalating categories (mild nomophobia = 21–59 of total NMP-Q score, moderate nomophobia = 60–99 of NMP-Q, and severe nomophobia >= 100 of NMP-Q). The independent variables in the model were the self-esteem categories (low vs. normal/high), the effect of phone use on academic performance, and the sociodemographic characteristics of the students such as gender, age, education, working status, residency, nationality, and parents’ education. It should be mentioned that Rosenberg’s scale focuses primarily on determining low (RSES < 15) and normal values (RSES =16–25) of self-esteem. Therefore, the binary scale <15= low and ≥ 15 = normal/high of self-esteem was used. The proportional odds assumption for the ordinal model was checked through the parallel lines test and graphically through cumulative percentage plots [22]. 

Results are presented as cumulative odds ratios (Cum OR) along with their corresponding 95% confidence intervals (95% CI). In the first run, the cumulative odds ratios for low vs. normal/high self-esteem participants were estimated as unadjusted by a univariate model. Then, they were adjusted for participants’ sociodemographic variables and the effect of mobile use on academic performance by a multivariate ordinal model. All statistical calculations were performed using SPSS version 28 statistical software (IBM SPSS, Armonk, NY, USA).

## 3. Results

The prevalence of mild level of nomophobia was 21.7%, of moderate level 59.6%, and of severe level 18.7%. Regarding self-esteem levels, 18.7% of the students showed low self-esteem, while the rest showed normal/high levels. Women, age group 18–20 years, non-working participants, and those whose fathers had no university education appeared to have higher levels of nomophobia (*p* values for linear trend ≤ 0.05). In addition, working participants showed normal/high self-esteem levels more often than non-working ones (86.9% vs. 79.4%, *p* value = 0.008) (Table 1).

Almost all students (93.6%) had a web connection in their smartphone, while they tended to show higher rates of connection as their degree of nomophobia increased (*p* value for linear trend < 0.001). Concerning students with severe nomophobia, 62.1% were checking their phone every 10 minutes, while 54.3% reported a negative impact on their academic performance. The corresponding percentages for mobile checking in participants with moderate and mild nomophobia were 36.9% and 19.1%, respectively (*p* value for linear trend < 0.001), while the reported negative impacts for academic performance were 50.5% for moderate and 42.2% for mild nomophobia students (*p* value = 0.009). In addition, severe nomophobia students cite all reasons for using the mobile phone in higher percentages than the rest (communication with family and friends 93.9%, mail 86.4%, lessons 86.9%, social media 92.9%, camera 84.3%, and web-based information 79.8%). Correspondingly, students who exhibited severe nomophobia report the use of their mobile phone in all daily activities at much higher percentages compared to the mild/moderate nomophobia ones (during eating 56.9%, lessons 56.6%, driving 6.6%, while being with company 55.6%, in public transports 85.9%, and during their alone time 95.7%).

Regarding self-estimation scales, students with low self-esteem were checking their smartphone more often than the rest (46.0% vs. 35.8% for every 10 minutes checking). Of the low self-esteem students, 68.7% felt that their academic performance was negatively affected (versus 45% of the normal/high). Concerning the reasons for mobile use, low self-esteem participants systematically reported lower rates for all reasons of use, compared to normal/high ones, with the only significant difference however regarding communication with family/friends (86.9% vs. 92.3%). On the contrary, low self-esteem participants reported the use of their mobile phone in all daily activities at higher rates compared to the normal/high self-esteem ones. However, statistically significant differences are observed in use while eating (48.2% vs. 34%), during lessons (53% vs. 40.7%), and while being with company (41.4% vs. 29.5%) (Table 2).

In terms of social networking, as was expected, participants with severe nomophobia made more use of the smartphone in their social life compared to others; they made more phone calls (about 8 calls/day), had more network friends (mean 1218 friends) and followers (mean 780 followers), and spent more hours per day on their smartphone (8.4 h/day). On the other hand, those with low self-esteem used their smartphone for more hours per day (7.5 h/day vs. 6.5 of the normal/high ones); however, compared to normal/high self-esteem participants, they made fewer calls (mean 5.6 vs. 6.6 calls/day), had fewer friends (mean 781 vs. 1063 friends), and had fewer followers (mean 520 vs. 651 followers) (Table 3).

To explore the relationship between self-esteem and nomophobia, unadjusted and adjusted cumulative odds ratios for higher nomophobia between low and normal/high self-esteem participants were estimated, using simple and multivariate ordinal models. Unadjusted analysis showed that for participants with low self-esteem, the cumulative odds ratio of exhibiting a higher level of nomophobia was more than double compared to the rest (Cum OR: 2.14; 95% CI: 1.57–2.91). After adjustment for potential sociodemographic confounders, the odds ratios remained statistically significant (adj Cum OR: 1.99; 95% CI: 1.45–2.73). Additionally, women and those whose father had no university education showed a higher risk of experiencing a greater level of nomophobia (adj Cum OR: 1.56; 95% CI: 1.19–2.04 and adj Cum OR: 1.44; 95% CI: 1.10–1.89, respectively) (Table 4). 

## 4. Discussion

### 4.1. Nomophobia and Self-Esteem Levels

Due to the emergence of nomophobia as an addiction in recent years, only a few studies so far have examined the association between nomophobia and self-esteem. According to the findings of the current study, almost 60% of the university students exhibited moderate and 18.7% a severe level of nomophobia. Similarly, a moderate level of nomophobia seems to prevail among university students [23,24,25], while they also appear to be more susceptible to severe nomophobia compared to the general population (25.46% vs. 21%) [26]. Concerning self-esteem levels, 18.7% of the university students showed low self-esteem and 81.3% showed normal/high levels, which is in accordance with the results of similar investigations [27,28].

### 4.2. Sociodemographic Characteristics

Regarding sociodemographic characteristics, females, those aged 18 to 20 years, non-working students, and those whose father had no university education were more likely to show severe levels of nomophobia than males, older age groups, working students, and those having a father with a university education, correspondingly. Previous studies have shown that female students were more susceptible to nomophobic behaviors [23,29], while younger college students in India (<20 years) had a higher prevalence of severe nomophobia (compared to older ages, ≥20 years) [30]. Additionally, a recent systematic review revealed that women and younger people were more prone to nomophobia [31]. Women seem to be attached to their smartphones to a greater extent, while younger people, due to the fact that they have integrated technology into their daily life, are more smartphone-dependent [32]. Moreover, Gnardellis et al. (2022) reported that all nomophobia subscales were adversely associated with the father’s higher education [20]. 

In regard to self-esteem, working participants showed generally higher levels of self-esteem. The results of a study conducted in Turkey revealed a positive correlation between self-esteem and working participants’ income level. It can be assumed that the availability of an income increases confidence and has a positive effect on self-esteem [33].

### 4.3. Nomophobia, Self-Esteem and Smartphone Use

A positive association between web connection and higher nomophobia levels was also observed. The availability of internet access is considered as one of the most important factors leading to smartphone addiction [34]. Furthermore, participants with severe nomophobia reported higher rates of smartphone use for social media, lessons, mail, camera, web-based information, and communication with family and friends. A recent study conducted in Malaysia revealed that a high percentage of participants (92%) used their smartphones for social media, receiving information (91.5%), and calling and exchanging SMS (87.6%) [35]. In the same line, nomophobia was positively associated with individuals using social media more often, exchanging more messages, and checking notifications more frequently [36].

The frequent smartphone checking was observed to a greater extent among students with severe nomophobia and low self-esteem. Recent findings showed that participants with severe levels of nomophobia check their smartphones more frequently than those with other nomophobia levels [19]. The results of a recent study indicated a substantial difference between the level of nomophobia and frequency of smartphone daily checking. It was observed that college students who checked their smartphones more than 50 times during the day had a higher nomophobia level than those who checked them less often [37]. 

In addition, those who had severe nomophobia or low self-esteem used their smartphones more frequently than their counterparts in daily activities (i.e., while eating, during class, or when socializing). The findings of another study revealed that participants used their smartphones while eating (40.9%), during lessons (48%), while driving (48%), while being with friends (52.8%), on public transportation (82.7%), and when they were alone (93.7%). Even though the direct relationship between daily use and the levels of nomophobia was not examined, a close connection seems to exist [38].

### 4.4. Nomophobia, Self-Esteem and Social Networking

Regarding social networking, it was revealed that participants with severe nomophobia made more phone calls/day, had more network friends and followers, and used their smartphones for more hours per day. In a recent study, students who used their phones primarily for social networking and messaging demonstrated a substantial risk of nomophobia [39], while another study claimed that the smartphone overuse for social networking increased the level of nomophobia [40].

As far as self-esteem and social networking are concerned, it was found that participants with low self-esteem used their smartphones for more hours per day, and those with normal/high self-esteem made more phone calls and had more network friends. However, it was observed that people with low self-esteem preferred calls, messages, emails, and social networking in order to communicate with other people [41].

### 4.5. Nomophobia, Self-Esteem and Academic Performance

In regard to academic performance, participants with severe nomophobia or low self-esteem reported a negative impact of smartphone excessive use on their academic achievement. A possible explanation of this double relationship is the low self-esteem per se which can create both the high nomophobia and the fear of low academic performance. Several studies manifested that nomophobia works at the expense of students’ academic performance [42,43,44], while others observed a significant positive association between self-esteem and academic performance [45]. However, the results of a recent study showed no statistically significant relationship between undergraduate pharmacy students’ self-esteem levels and academic attainment [46].

### 4.6. Relationship between Nomophobia and Self-Esteem

It was interesting to observe that participants with low self-esteem were twice as likely to demonstrate nomophobia compared to those with normal/high levels. This finding is in accordance with other studies which found a significant negative correlation between self-esteem and nomophobia in university students [1,14,19,47]. On the other hand, Ozdemir et al. (2018) indicated that nomophobia was positively associated with self-esteem [8], while in another research study, no significant association was evident among nomophobia and self-esteem [28]. 

Finally, participants who have a father without a university education showed a higher risk of developing greater levels of nomophobia. However, in another study which investigated nomophobia levels in secondary school students, it was noted that when their father’s educational level was high, the mean nomophobia score was low [48]. Due to the lack of such evidence, more research is necessary to establish the link between the father’s educational level and nomophobia levels. 

Since the study was limited to students from one university in the Attica prefecture, generalizing the findings is somewhat difficult. However, in terms of both student enrolment and the number of faculties, the specific university is the third largest in Greece. Regarding gender, even though women were overrepresented related to men, both samples (women and men) were sufficiently large, and the composition of the total sample did not affect the final relationships. Moreover, all relationships were adjusted for the confounding effects of gender. Additionally, the results shed light on a very important and crucial issue.

## 5. Conclusions

Nomophobic symptoms are very prevalent in university students and often affect their academic performance. Self-esteem and nomophobia were closely connected. Programs for the prevention of nomophobia should not only focus on individuals’ smartphone adhesion, but also on the psychological pathways that may lead to nomophobic behaviors. In a broader context of investigating the relationship between nomophobia and self-esteem, an evaluation of other parameters such as depression, anxiety, and stress would be very interesting. In a future work, we will study this complex relationship, i.e., between nomophobia and self-esteem, taking into account the effects of depression, anxiety, and stress.

## Figures and Tables

**Table 1 ijerph-20-02929-t001:** Sociodemographic characteristics of study subjects by NMP and self-esteem categories.

			Nomophobia			*χ*^2^ for Linearity *p*-Value	Self-Esteem		*χ*^2^ Test *p*-Value
			Mild	Moderate	Severe		Low	Normal/High	
	N	%	*n*_1_ = 230 21.7	*n*_2_ = 632 59.6	*n*_3_ = 198 18.7		*n*_1_ = 198 18.7	*n*_2_ = 862 81.3	
Gender									
Women	752	70.9	19.3	60.1	20.6	<0.001	18.4	81.6	0.733
Men	308	29.1	27.6	58.4	14.0		19.5	80.5	
Age groups									
18–20	576	54.3	19.8	59.4	20.8	0.023	20.3	79.7	0.159
21+	484	45.7	24.0	59.9	16.1		16.7	83.3	
Work									
No	785	74.1	20.3	59.2	20.5	0.005	20.6	79.4	0.008
Yes	275	25.9	25.8	60.7	13.5		13.1	86.9	
Residency									
With parents	828	78.1	20.9	61.4	17.8	0.907	19.6	80.4	0.193
Alone	232	21.9	24.6	53.4	22.0		15.5	84.5	
Nationality									
Greek	998	94.2	21.5	59.8	18.6	0.950	18.5	81.5	0.814
Other	58	5.5	24.1	55.2	20.7		20.7	79.3	
Father Education									
Other	687	64.8	19.4	60.0	20.6	0.003	19.5	80.5	0.393
University	373	35.2	26.0	59.0	15.0				
Mother Education									
Other	571	53.9	21.7	58.0	20.3	0.310	19.8	80.2	0.356
University	489	46.1	19.7	61.6	18.7		17.4	82.6	

Note: *n* = number of subjects (*n*_1_, *n*_2_, *n*_3_)

**Table 2 ijerph-20-02929-t002:** Mobile phone use in percentages according to NMP and self-esteem categories.

		Nomophobia Categories			*χ*^2^ for Linearity *p*-Value	Self-Esteem		*χ*^2^ Test *p*-Value
%	Total	Mild	Moderate	Severe		Low	Normal /High	
Web connection in phone	93.6	87.8	94.6	97.0	<0.001	90.9	94.2	0.102
Checking								
Up to 10 min	37.7	19.1	36.9	62.1	<0.001	46.0	35.8	0.004
20 min	20.1	16.5	21.2	20.7		21.7	19.7	
30 min	17.1	20.0	18.7	8.6		9.6	18.8	
>30 min	25.1	44.3	23.3	8.6		22.7	25.6	
Possession of 2nd mobile phone	14.6	16.1	13.8	15.7	0.857	17.2	14.0	0.310
Cost of mobile phone								
<200 euro	44.3	59.6	43.8	28.3	<0.001	49.5	43.2	0.039
200–400 euro	32.2	25.2	33.9	34.8		33.8	31.8	
>400 euro	23.5	15.2	22.3	36.9		16.7	25.1	
Smartphone affects academic performance negatively	49.4	42.2	50.5	54.3	0.009	68.7	45.0	<0.001
Reasons to use smartphone								
Communication with family/friends	91.3	87.8	91.8	93.9	0.023	86.9	92.3	0.020
Mail	78.9	70.0	79.7	86.4	<0.001	78.3	79.0	0.899
Lessons	83.9	79.1	84.7	86.9	0.027	79.8	84.8	0.105
Social Media	80.9	65.2	82.9	92.9	<0.001	77.3	81.8	0.174
Camera	72.2	60.4	72.6	84.3	<0.001	67.7	73.2	0.140
Web-based information	69.9	55.7	72.0	79.8	<0.001	65.7	70.9	0.174
When he/she uses smartphone								
Use/during eating	36.6	22.6	35.4	56.9	<0.001	48.2	34.0	<0.001
Use/during lessons	43.0	28.3	44.1	56.6	<0.001	53.0	40.7	0.002
Use/during driving	2.5	2.2	1.4	6.6	0.007	3.5	2.3	0.466
Use/while being with company	31.7	12.6	31.2	55.6	<0.001	41.4	29.5	0.002
Use/in public transportation	80.8	67.4	84.2	85.9	<0.001	84.8	79.9	0.137
Use/when he/she is alone	93.9	86.5	95.7	96.5	<0.001	94.4	93.7	0.833

**Table 3 ijerph-20-02929-t003:** Mobile phone use by NMP and self-esteem categories.

			Nomophobia Categories							Self-Esteem				
	Total		Mild		Moderate		Severe		*p*-Value ^1^	Low		Normal/High		*p*-Value ^1^
	Mean	Median	Mean	Median	Mean	Median	Mean	Median		Mean	Median	Mean	Median	
Calls/day	6.5	5	6.7	5	5.9	5	7.9	6	0.004	5.6	4	6.6	5	<0.001
Messages/day	23.6	20	23.2	20	23.4	20	24.6	20	0.564	21.9	20	23.9	20	0.128
Emails/day	7.8	7	7.5	7	7.8	7	8.3	7	0.069	7.9	7	7.8	7	0.564
Friends ^2^	1010	700	969	508.5	960	618	1218	950	<0.001	781	567.5	1063	717.5	<0.001
Followers ^3^	626	449.5	498	335	625	450	780	551	<0.001	520	400	651	450	0.073
Phone use h/day	6.7	6	5.1	5	6.7	6	8.4	8	<0.001	7.5	7	6.5	6	0.001
Computer use h/week	20	15	21.7	15	20	15	18.8	10	0.113	20.2	15	20	15	0.699

^1^ Kruskal–Wallis nonparametric test; ^2^ (Fb, MSN, games); ^3^ (Fb, Insta, Twitter).

**Table 4 ijerph-20-02929-t004:** Cumulative odds ratios derived from ordinal logistic regressions with NMP categories as dependent variable.

	Univariate Ordinal Model with NMP Categories as Dependent Variable		
	Cumulative OR	95%CI	*p*-Value
Self-esteem			
Low vs. Normal/High	2.14	1.57–2.91	<0.001
	**Multivariate ordinal model with NMP categories as dependent variable**		
	**Adj Cumulative OR**	**95%CI**	***p*-Value**
Self-esteem			
Low vs. Normal/High	1.99	1.45–2.73	<0.001
Smartphone affects academic performance negatively	1.22	0.95–1.56	0.118
Gender			
Women (vs. Men)	1.56	1.19–2.04	0.001
Age groups			
18–20 (vs. 21+)	1.22	0.95–1.58	0.124
Work			
No (vs. Yes)	1.27	0.95–1.69	0.105
Residency			
With parents (vs. Alone)	0.92	0.68–1.23	0.563
Nationality			
Greek (vs. other)	1.10	0.65–1.87	0.726
Father Education			
Other (vs. University)	1.44	1.10–1.89	0.008
Mother Education			
Other (vs. University)	1.02	0.79–1.32	0.880

## Data Availability

Data are available upon request.

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
