# Peer review of "Nomophobia and Self-Esteem: A Cross Sectional Study in Greek University Students"

_ijerph, 2023, doi:10.3390/ijerph20042929_

Round 1

Reviewer 1 Report

Comments to the Authors

Major Comments

1.2. Rosenberg Self Esteem Scale (RSES)

I suggest that the authors should report Cronbach's alpha for Rosenberg Self Esteem Scale as they have already done for NMP-Q.

Line 130

Women are overrepresented in the research sample (70.9%). The authors should report if this has any impact on the interpretation of the findings, given that gender seems to be linked to the NMP-Q categories, and they should possibly include it in the research limitations.

Lines 101 & 102

“….. values less than 15 indicates low self-esteem, a score of 15-25 indicates normal, and a score of more than 25 indicates high.”

The authors should explain why, in all statistical analyses (Tables 1, 2, 3 and 4), they treat Rosenberg Self Esteem Scale as a binary variable (low vs normal/high), while it is made up of three categories.

Minor Comments

Lines 86 & 87

“The nomophobia total score is between 20 and 140 (<20 absence, …..).”

It is not clear why the authors state that values of the total score less than 20 correspond to the absence of nomophobia while the minimum possible score is 20.

Tables 1 & 4

Replace “affects academic performance” by “affects negatively academic performance” in accordance with what has been reported in line 141  “… reported a negative impact on their academic performance”.

Line 68

“…. has been recently gained..”

Remove the word “been” .

Line 118
“ … resistency”.
Replace by “residency”.

Correct mistakes in ref. #5: double f is needed for “efects” and “diferences”.

Author Response

We would like to thank the Reviewer very much for the time spent on our work, as well as the useful comments made that helped us to improve the presentation of our findings.

Major Comments

 1.2. Rosenberg Self Esteem Scale (RSES)

I suggest that the authors should report Cronbach's alpha for Rosenberg Self Esteem Scale as they have already done for NMP-Q.

Reply: Cronbach's alpha value (=0.81) was calculated for the self-esteem scale and is incorporated in the revised ms (pls see lines 110-111).

Line 130

Women are overrepresented in the research sample (70.9%). The authors should report if this has any impact on the interpretation of the findings, given that gender seems to be linked to the NMP-Q categories, and they should possibly include it in the research limitations.

Reply: The men in the study sample were 308 (29.1%) and the women 752 (70.9%). Because both samples are sufficiently large, (much larger than 30) the composition of the total sample with a higher representation of women over men does not affect the final relationships. After all, in the ordinal regression model, the calculated odds ratios are adjusted for the possible confounding effects of gender, since it is included as an independent variable in the model. Regardless of whether gender is significant in the ordinal regression, the adj cumulative odds ratio calculated for self-esteem is adjusted for the confounding effects of gender. (pls see Limitation Section in the revised ms).

Lines 101 & 102

“….. values less than 15 indicates low self-esteem, a score of 15-25 indicates normal, and a score of more than 25 indicates high.”

The authors should explain why, in all statistical analyses (Tables 1, 2, 3 and 4), they treat Rosenberg Self Esteem Scale as a binary variable (low vs normal/high), while it is made up of three categories.

Reply: Rosenberg's scale focuses primarily on determining low (RSES < 15) and normal values (RSES =16-25) of self-esteem. This is the main distinction of the scale and is the central focus of studies that use it as a self-esteem measurement tool. In addition, in our study we focused on low self-esteem because the dysfunctional smart phone use is mainly associated with low levels of self-esteem [References 14,15 of the article]. That’s why we used the scale with the two distinct limits of < 15 = low and ≥ 15 = normal/high. In separate analyses carried out (not included in the article), it emerged that self-esteem values > 25 correspond to severe nomophobia at a rate of 16.6%, while self-esteem values between 16 and 25 at an almost identical rate of 16.2%. Similar percentages were found for the level of mild nomophobia.  In all analyses performed, the results of self-esteem values for the interval [15-25] and >25 were similar. In addition, our results were more powerful with the binary coding of RSES while the proportional odds assumption was ensured in the ordinal regression model (pls see Data Analysis Section, in the revised ms).

Minor Comments

Lines 86 & 87

“The nomophobia total score is between 20 and 140 (<20 absence, …..).”

It is not clear why the authors state that values of the total score less than 20 correspond to the absence of nomophobia while the minimum possible score is 20.

 Reply: The reviewer is correct. On the nomophobia scale, absence = 20. By typo mistake it was written < 20. It was also corrected in the revised ms (pls see line 93).

Tables 1 & 4

Replace “affects academic performance” by “affects negatively academic performance” in accordance with what has been reported in line 141  “… reported a negative impact on their academic performance”.

Reply: The addition was made as requested in the revised ms (pls see tables 2 & 4).

Line 68

“…. has been recently gained..”

Remove the word “been” .

Reply: The word been has been removed (pls see line 68).

Line 118
“ … resistency”.
Replace by “residency”.

Reply: The word was replaced as requested in the revised ms (pls see line 125).

Correct mistakes in ref. #5: double f is needed for “efects” and “diferences”.

Reply: Mistakes were corrected in the revised ms (pls see reference #5).

Reviewer 2 Report

If possible, author can divide the participants in Rosenberg Self Esteem Scale, as low, normal and high self-esteem separately, rather than low self-esteem and clubbing normal/high self-esteem together for analysis.

Author Response

If possible, author can divide the participants in Rosenberg Self Esteem Scale, as low, normal and high self-esteem separately, rather than low self-esteem and clubbing normal/high self-esteem together for analysis.

Reply: We would like to thank the Reviewer very much for the time spent on our work and the comment made.

Rosenberg's scale focuses primarily on determining low (RSES < 15) and normal values (RSES =16-25) of self-esteem. This is the main distinction of the scale and is the central focus of studies that use it as a self-esteem measurement tool. In addition, in our study we focused on low self-esteem because the dysfunctional smart phone use is mainly associated with low levels of self-esteem [References 14,15 of the article]. That’s why we used the scale with the two distinct limits of < 15 = low and ≥ 15 = normal/high. In separate analyses carried out (not included in the article), it emerged that self-esteem values > 25 correspond to severe nomophobia at a rate of 16.6%, while self-esteem values between 16 and 25 at an almost identical rate of 16.2%. Similar percentages were found for the level of mild nomophobia.  In all analyses performed, the results of self-esteem values for the interval [15-25] and >25 were similar. In addition, our results were more powerful with the binary coding of RSES while the proportional odds assumption was ensured in the ordinal regression model (pls see Data Analysis Section, in the revised ms).

In support of the above, we provide Τable 4 of the regression analysis with independent variable the self-esteem scale in three categories, low, normal, high. As in the manuscript, unadjusted analysis showed that for participants with low self-esteem the odds ratio of exhibiting higher level of nomophobia was at least twice larger as the normal and high levels (Cum OR: 2.00 and 2.46 respectively). While after adjustment for potential sociodemographic confounders, the odds ratios of self-esteem remained significant with Cum OR for low vs normal and high 1.89 and 2.24 respectively. Moreover, the rest of the model parameters remain almost the same as in the original analysis of Table 4.

Cumulative odds ratios derived from ordinal logistic regressions with NMP categories as dependent variable

 Univariate ordinal model with NMP categories as dependent variable

Cumulative OR

95%CI

p value

Self-esteem

  Low vs Normal

2.00

1.45 -2.76

<0.001

  Low vs High

2.46

1.71-3.54

<0.001

Multivariate ordinal model with NMP categories as dependent variable

Adj Cumulative OR

95%CI

p value

Self-esteem

 Low vs Normal

1.89

    1.36-2.63

<0.001

    Low vs High

2.24

    1.54-3.25

<0.001

Smartphone affects academic performance negatively

1.20

    0.94-1.54

  0.148

Gender

Women (vs Men)

1.55

1.19-2.03

  0.001

Age groups

18-20 (vs 21+)

1.22

0.94-1.57

  0.132

Work

No (vs Yes)

1.26

1.11-1.69

  0.108

Residency

With parents (vs Alone)

 0.91

0.68-1.22

  0.539

Nationality

Greek (vs other)

1.10

0.65-1.87

  0.728

Father Education

Other (vs University)

   1.44

  1.10-1.89

     0.008

Mother Education

Other (vs University)

 1.03

0.79-1.33

   0.850

Reviewer 3 Report

Dear Authors,

Thank you for the opportunity to review an interesting article entitled: ‘Nomophobia and Self-Esteem: A Cross Sectional Study in Greek University Students’. The aim of the article is to analyse the relationship between nomophobia and self-esteem. The subject matter is interesting and extremely topical.

The strengths of the article presented for evaluation are the large sample size, the use of statistical analyses and socio-demographic variables rarely considered in the analysis of nomophobia. The citation of current literature is also a strength.

The reviewer's job, on the other hand, is to help improve the article so that it meets the highest possible standards of the journal, therefore I will focus on its weaknesses.

General comments

[1].  The structure of the "Material and methods" section needs to be improved. The sections "Subjects", "Study procedure", "Tools used", "Data analysis" should be distinguished here.

[2].  Text throughout the manuscript should be uniformly formatted.

[3].  The numbering of parts in the 'Materials and methods' section should be corrected.

Material and methods

[4].  Subjects should be described on the basis of the socio-demographic variables listed in lines 82-83.

Results

[5].  Information characterising the subjects should be described in the 'Material and methods' section - e.g. from lines 129-130.

[6].  Line 145 lacks a full stop at the end of the sentence.

References

[7].  DOI numbers should be completed in the bibliography.

Author Response

Thank you for the opportunity to review an interesting article entitled: ‘Nomophobia and Self-Esteem: A Cross Sectional Study in Greek University Students’. The aim of the article is to analyse the relationship between nomophobia and self-esteem. The subject matter is interesting and extremely topical.

The strengths of the article presented for evaluation are the large sample size, the use of statistical analyses and socio-demographic variables rarely considered in the analysis of nomophobia. The citation of current literature is also a strength.

The reviewer's job, on the other hand, is to help improve the article so that it meets the highest possible standards of the journal, therefore I will focus on its weaknesses.

Reply: We would like to thank the Reviewer very much for the time spent on our work, as well as the useful comments made that helped us to improve the ms.

General comments

[1].The structure of the "Material and methods" section needs to be improved. The sections "Subjects", "Study procedure", "Tools used", "Data analysis" should be distinguished here.

Reply: The structure of the "Material and methods" section has been improved (pls see "Material and methods" section of the revised ms).

[2]. Text throughout the manuscript should be uniformly formatted.

Reply: The reformatted has been done according to the journal's article template (pls see the revised ms).

[3].The numbering of parts in the 'Materials and methods' section should be corrected.

Reply: They were corrected (pls see the revised ms).

Material and methods

[4]. Subjects should be described on the basis of the socio-demographic variables listed in lines 82-83.

Reply: They were described in the revise ms (pls see "Material and methods" section of the revised ms).

Results

[5]. Information characterising the subjects should be described in the 'Material and methods' section - e.g. from lines 129-130.

Reply: The certain information has been added in the “Material and Methods” section of the revised ms (pls see "Material and methods" section of the revised ms).

[6]. Line 145 lacks a full stop at the end of the sentence.

Reply: The full stop has been added (pls see line 157).

References

[7]. DOI numbers should be completed in the bibliography.

Reply: DOI numbers have been added in the revise ms. Only in reference 30 doi number does not exist (pls see references).